**Eco-evolutionary Modelling of Global Vegetation Dynamics and the Impact of CO$_2$**
**during the late Quaternary: Insights from Contrasting Periods**
Jierong Zhao[1], Boya Zhou[2], Sandy P. Harrison[1,*], Iain Colin Prentice[2]
[1] Department of Geography and Environmental Science, University of Reading, Whiteknights,
Reading, RG6 6AB, UK
[2] Georgina Mace Centre for the Living Planet, Department of Life Sciences, Imperial College
London, Silwood Park Campus, Buckhurst Road, Ascot, SL5 7PY, UK
* Corresponding author: s.p.harrison@reading.ac.uk
*Ms for: Earth System Dynamics*
**Abstract**
Changes in climate have had major impacts on global vegetation during the Quaternary.
However, variations in CO$_2$ levels also play a role in shaping vegetation dynamics by
influencing plant productivity and water-use efficiency, and consequently the relative
competitive success of the C$_3$ and C$_4$ photosynthetic pathways. We use an eco-evolutionary
optimality (EEO) based modelling approach to examine the impacts of climate fluctuations and
CO$_2$-induced alterations on gross primary production (GPP). We considered two contrasting
periods, the Last Glacial Maximum (LGM, 21,000 years before present) and the mid-Holocene
(MH, 6,000 years before present) and compared both to pre-industrial conditions (PI). The
LGM, characterised by generally colder and drier climate, had a CO$_2$ level close to the
minimum for effective C$_3$ plant operation. In contrast, the MH had warmer summers and
increased monsoonal rainfall in the northern hemisphere, although with a CO$_2$ level still below
PI. We simulated vegetation primary production at the LGM and the MH compared to the PI
baseline using a light-use efficiency model that simulates GPP coupled to an EEO model that
simulates leaf area index (LAI) and C$_3$/C$_4$ competition. We found that low CO$_2$ at the LGM
was nearly as important as climate in reducing tree cover, increasing the abundance of C$_4$ plants
and lowering GPP. Global GPP in the MH was similar to the PI (although greater than the
LGM), also reflecting CO$_2$ constraints on plant growth despite the positive impacts of warmer
and/or wetter climates experienced in the northern hemisphere and tropical regions. These
results emphasise the importance of taking account of impacts of changing CO$_2$ levels on plant
growth to model ecosystem changes.

## 1 Introduction

Vegetation regulates the exchanges of energy, water, and carbon dioxide between the land and the atmosphere (Williams and Torn, 2015; Forzieri et al., 2020; Hoek van Dijke et al., 2020). Gross primary production (GPP), defined as the carbon uptake by vegetation through photosynthesis at the ecosystem scale, determines the extent to which the terrestrial biosphere can mitigate $CO_2$ emissions (Bonan, 2008; Zeng et al., 2017; Chen et al., 2019). There is a tight coupling between $CO_2$ uptake and water loss via stomata, such that when ambient $CO_2$ is high water-use efficiency (the amount of water required for transpiration to achieve a unit of $CO_2$ assimilation) is also high (Medlyn et al., 2017). Recent global greening trends are thought to reflect both changes in climate, particularly warming at high latitudes, and the effect of increasing $CO_2$ on water-use efficiency (Cai and Prentice, 2020; Piao et al., 2020). However, there is still uncertainty about the relative importance of these two effects on recent changes in global GPP, in part because recent climate changes have been largely driven by the increase in $CO_2$.

Past climate states provide opportunities to examine the role of climate and $CO_2$ in modulating GPP when there is a greater de-coupling between changes in $CO_2$ and climate. The contrast between glacial and interglacial states during the Late Quaternary offers an ideal opportunity to separate the impact of these two factors on vegetation. Glacial-interglacial shifts in climate are largely driven by changes in orbital configuration which resulted in changes in the seasonal and latitudinal patterns of incoming solar radiation (Berger, 1978; Berger and Loutre, 1991). The Last Glacial Maximum (LGM), ca 21,000 years ago, had an orbital configuration similar to the present but was characterised by the presence of large continental ice sheets and generally colder and drier climates (Kageyama et al., 2021). The $CO_2$ level was ca 190 ppm, which is close to the minimum for effective $C_3$ plant operation (Gerhart and Ward, 2010). The mid-Holocene (MH), ca 6000 years ago, was characterised by a significantly different seasonal and latitudinal distribution of incoming solar radiation (a result of changes in obliquity and precession) which affected light availability for photosynthesis and produced warmer summers in the northern hemisphere and wetter conditions in the sub-tropics (Brierley et al., 2020). However, ambient $CO_2$ was only ca 264 ppm (Otto-Bleisner et al., 2017), somewhat lower than the pre-industrial (PI) period (285 ppm) and considerably lower than today.

Three sets of factors could potentially impact vegetation productivity changes between the LGM, MH and pre-industrial periods: changes in climate, atmospheric CO2 and solar radiation. Several published studies have simulated LGM climate impacts on vegetation (and/or fire, interacting with vegetation), with – or without – consideration of the additional physiological effects of low CO2 on plants (Levis et al. 1999, Harrison and Prentice 2003, Martin Calvo et al. 2014). Other studies have performed factorial experiments to more formally separate the effects of climate and CO2 (Woillez et al. 2011, O'ishi & Abe-Ouchi 2013, Claussen et al. 2013, Martin Calvo & Prentice 2015, Chen et al. 2019, Haas et al. 2023).

Comparison among these studies of LGM-to-recent primary production shifts is approximate at best because they have used different climate models and experimental protocols. Some have used pre-industrial conditions as a reference; others, modern (higher-CO2) conditions. However, they all have used land ecosystem models based on the plant functional type (PFT) concept. Uncertainty in the delimitation of PFTs and the parameter values assigned to them is endemic to this type of model, as variation of quantitative traits within PFTs in the real world is generally much larger than variation between them (Kattge et al., 2011). In some cases, the model PFT representation has resulted in an unrealistic simulation of LGM vegetation patterns

(e.g. Woillez et al. 2011). Here we use the P model (Stocker et al. 2020), which accounts for
acclimation and adaptation to environment independently of PFTs on the basis of universal
eco-evolutionary optimality (EEO) hypotheses. The P model has been subject to extensive
evaluation against worldwide data from eddy covariance flux towers across all biomes. We
include an extension of the P model which simulates foliage cover and its seasonal cycle – also
independently of PFTs. This extended model has been shown to reproduce foliage amounts
and seasonal dynamics more accurately than any state-of-the-art vegetation model (Zhou et al.,
2025). We use a simple process-based scheme to represent the relative competitive success of
$C_3$ versus C4 plants, which has been validated against worldwide soil carbon stable isotope
data (Lavergne et al., 2024). This combination of three independently tested, PFT-independent
modelling components enables us, for the first time, to apply an EEO-based approach to
simulate LGM and recent vegetation function in a globally uniform way
There has been some work on the implications of MH climate for biome distributions (e.g.
Kaplan et al., 2003; Wohlfahrt et al., 2008) but little consideration of the impacts of climate
and CO2 on global productivity changes since the MH (Foley, 1994; François et al. 1999).
Here, we use the same consistent methodology that we apply to the LGM to estimate MH-to-
pre-industrial changes in global GPP. Our analysis includes the effect of changes in the light
regime, which are a necessary consequence of changes in the seasonal and latitudinal
distribution of insolation due to orbital forcing, as well as changes in cloud cover linked to
monsoon shifts.
EEO-based modelling approaches provide parameter-sparse representations of plant and
vegetation processes, thus considerably reducing uncertainties due to model parameterisation
(Harrison et al, 2021). They have been shown to perform as well or better than more complex
models under recent conditions (Cai et al., 2025; Zhou et al., 2025) and thus can provide a
robust way of modelling vegetation changes under different climate regimes. We use a series
of counter-factual experiments to examine the magnitude of changes due to individual drivers
(climate parameters, solar radiation and $CO_2$) on the simulated GPP and to determine the
regions where specific factors are most influential.
**2 Methods**
**2.1 Modelling Scheme**
We simulated vegetation changes at the LGM and the MH compared to the pre-industrial (PI)
state using a sequence of linked models that predict GPP, leaf area index (LAI) and $C_3$/$C_4$
competition based on EEO theory (Fig. 1). We first simulate potential GPP (equivalent to leaf-
level photosynthesis) for $C_3$ and $C_4$ plants independently. These estimates are used to derive
total potential GPP allowing for competition between $C_3$ and $C_4$ plants. Potential GPP is
converted to actual GPP using a model that simulates the seasonal cycle of leaf area index
(LAI), which is converted to the fraction of absorbed photosynthetically active radiation
(fAPAR) using using Beer's law. Finally, we use a soil water balance model to calculate soil
moisture, then take account of the impact of low soil mosture on GPP using an empirical
correction.
The P model (Wang et al., 2017, Stocker et al., 2020) is a light-use efficiency model that
simulates GPP. It uses the Farquhar-von Caemmerer-Berry photosynthesis model (Farquhar et
al., 1980) for instantaneous biochemical processes combined with two EEO hypotheses
describing photosynthetic acclimation, the 'coordination' and 'least- cost' hypotheses (Prentice
et al., 2014, Wang et al., 2017), to account for the spatial and temporal acclimation of
carboxylation and stomatal conductance to environmental variations at weekly to monthly time
scales. Although the P model simulates both $C_3$ and $C_4$ photosynthesis, it does not need to make
any other distinctions between plant functional types. The required inputs to the model (Fig. 1)
are air temperature (°C), vapour pressure deficit (VPD, Pa) derived from relative humidity, air
pressure (Pa) (to account for the effect of elevation on photosynthesis, incident photosynthetic
photon flux density (PPFD, $\mu mol\ m^{-2}\ s^{-1}$) estimated from short wave solar radiation, and
ambient $CO_2$ concentration. The P model has been extensively validated and shown to predict
the geographic patterns and seasonal cycles of GPP under modern conditions successfully
(Wang et al., 2017; Stocker et al., 2020). Furthermore, it correctly predicts related physiological
characteristics, including the global pattern of the maximum carboxylation ($V_{cmax}$) rate in
relation to gradients in PPFD, temperature and VPD (Smith et al., 2019), the seasonal variation
of $V_{cmax}$ in different biomes (Jiang et al., 2020), its response to atmospheric $CO_2$ (Smith and
Keenan, 2020), and the variation of photosynthetic traits along elevational gradients (Peng et
al., 2020). The responses of photosynthetic properties to enhanced $CO_2$ as simulated by the P
model have been validated against both Free Air Carbon dioxide Enrichment (FACE)
experiments (Wang et al., 2017) and controlled-environment experiments (Smith and Keenan,
2020). Moreover, the model's implied response of photosynthetic capacity to $CO_2$ has been
validated by measurements on plants experimentally grown at low (160 ppm) $CO_2$ (Harrison
et al., 2021).

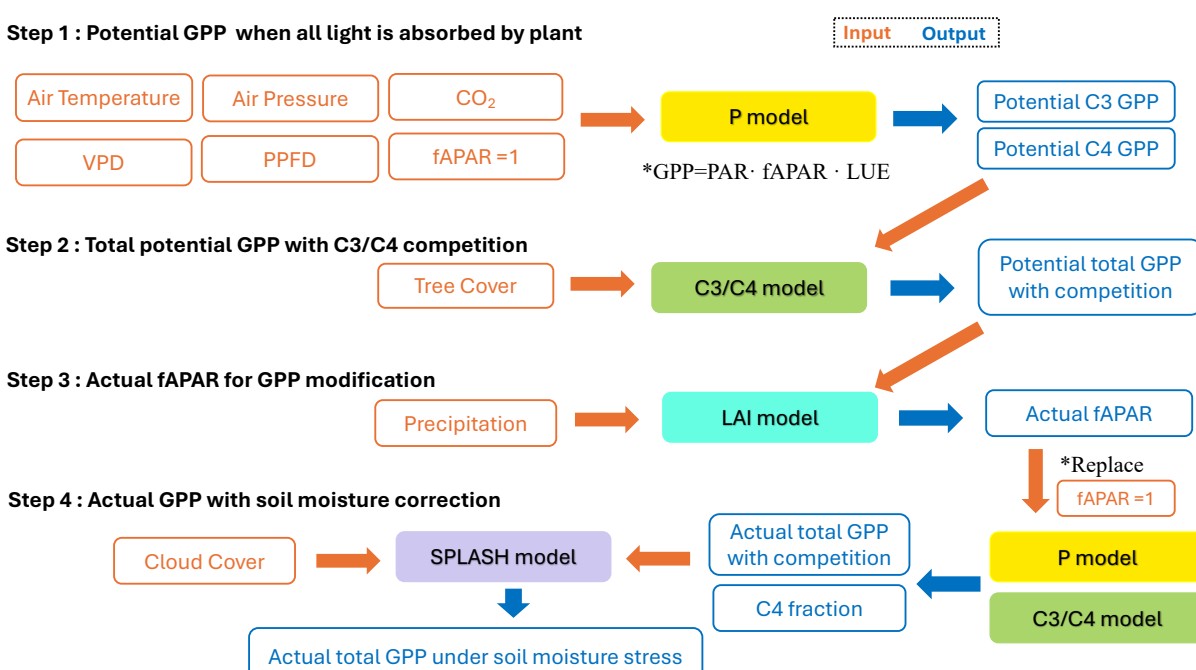

**Figure 1**: *Flowchart showing the steps in the modelling procedure.*

The P model first simulates potential GPP for $C_3$ and $C_4$ plants separately, without
consideration of competition between them (Figure 1). These estimates are fed into a simple
model of $C_3/C_4$ competition based on the P model (Lavergne et al., 2024). The principle of the
$C_3/C_4$ model is as follows. $C_4$ plants (mainly grasses, some shrubs) have a higher rate of
photosynthesis under hot and/or dry conditions, especially under low $CO_2$, which reduces $C_3$
photosynthesis. On the other hand, $C_4$ plants can only become dominant or co-dominant if tree

cover is limited, because (C$_3$) trees have the advantage in competition for light: C$_3$ trees can outcompete C$_4$ grasses through shading even where the C$_4$ pathway would yield higher rates of photosynthesis. The relative photosynthetic advantage of C$_4$ plants is estimated in the model as the difference between the monthly potential GPP for C$_3$ and C$_4$ plants, summed over the year. The C$_4$ share of total GPP was then estimated based on a logistic relationship between the model-estimated C$_4$ relative advantage and observed C$_4$ abundance. An additional function relates the proportion of GPP from trees to total potential GPP based on the relationship between annual mean percentage tree cover and the simulated annual GPP of C$_3$ plants. Thus, tree cover is an additional required input to the competition model (Figure 1). The competition model also enforces a minimum temperature threshold to define conditions under which C$_4$ plants cannot grow, where this limit is set to a minimum temperature of the coldest month of –24° based on experimental data. The competition model has been shown to reproduce global patterns in the relative abundance of C$_3$/C$_4$ plants as well as the observed rate of $\Delta^{13}C$ in recent decades, as shown by independent atmospheric estimates (Lavergne et al., 2020).

To convert potential GPP to actual GPP, we used an LAI model (Figure 1) that predicts the seasonal cycle of LAI based on environmental conditions and the P-model estimates of potential GPP, i.e. the GPP predicted when the fraction of absorbed photosynthetically active radiation, fAPAR, is set to 1 (Zhou et al., 2025). This model is based on the EEO hypothesis that seasonal variations in LAI are coordinated with variations in potential GPP because leaves are displayed at (or close to) the time when they are able to be most productive. A seasonal maximum LAI model was embedded in this model to provide an upper limit to the seasonal LAI predictions (Zhu et al., 2022; Cai et al., 2025). The calculation of seasonal maximum LAI incorporates a water-carbon trade-off and is defined as the lesser of an energy-limited and a water-limited estimate (Zhu et al., 2022; Cai et al., 2025). The model assumes that, under energy limitation, plants allocate carbon to leaves to maximize GPP after accounting for the costs of leaf construction and maintenance, including the costs of obtaining water and nutrients. This leads to a clear optimum because investing in leaf tissue yields diminishing returns due to mutual leaf shading. Under water limitation, the model assumes that plants adjust their rooting behaviour to extract a portion of annual precipitation from the soil, irrespective of its distribution throughout the year, and allocate carbon to leaves so that all this water is transpired, thereby maximizing GPP. There are inherent delays between the steady-state LAI and the real-time dynamic LAI because photosynthetic and phenological processes do not respond instantaneously to weather fluctuations: the allocation of photosynthate to leaves can take days to months. Thus, the seasonal dynamics of LAI were calculated using a moving average to represent the time lag between allocation to leaves and modelled steady-state LAI (Zhou et al., 2025). The model has been shown to capture observed LAI dynamics across all biomes at different temporal scales (weekly, seasonal, annual and interannual variability) both at individual eddy-covariance flux measurement sites and when compared to satellite-derived LAI (Zhou et al., 2025). Furthermore, it predicts both the multi-year average LAI and the annual trends in LAI better than the biosphere models used in the Trends and Drivers of Terrestrial Sources and Sinks of Carbon Dioxide (TRENDY) project (Zhou et al., 2025). The seasonal cycle of fAPAR is calculated from the seasonal cycle of LAI using Beer's law (Swinehart, 1962) and this is then used to calculate seasonal changes in actual GPP.

Finally, we apply an empirical soil moisture correction ($\beta(\theta)$: Stocker et al., 2020) to account for the additional impact of soil moisture stress on GPP. This has the form of a quadratic expression applied when soil moisture is below a given threshold value, where the sensitivity of this relationship varies depending on aridity such that the decline in β(θ) with drying soils is steep in dry climates and less steep in wetter climates. The soil moisture stress function was

developed by comparing simulations of GPP with flux-tower data at sites across a range of climatic aridity (Stocker et al., 2020). Soil moisture was calculated using the Simple Process-Led Algorithms for Simulating Habitats (SPLASH) model (Davis et al., 2017), which is a generic soil water accounting model in which daily losses depend on potential evaporation, reduced in proportion to relative soil water content.

## 2.2. Derivation of LGM, MH and PI climate and vegetation inputs

We use LGM, MH and pre-industrial (PI) climate simulations (Supplementary Table 1) run using the low-resolution version of the Max Planck Institute Earth System Model (MPI-ESM1.2-LR; Mauritsen et al., 2019; doi:10.22033/ESGF/CMIP6.6642) made as part of the fourth phase of the Palaeoclimate Modelling Intercomparison Project (PMIP4; Kageyama et al., 2017; Otto-Bleisner et al., 2019). This model is amongst the best performing of the PMIP models when evaluated using reconstructions of land and ocean climates (Brierley et al., 2020; Kageyama et al., 2021) and uniquely has archived all the necessary climate and vegetation outputs needed to run the EEO-based models (Fig. 1). The experiments were run following the PMIP4 protocols for each time period (Kageyama et al., 2017; Otto-Bleisner et al., 2019). The PI experiment was run for 1000 years using modern ice sheet and land-sea configurations and a $CO_2$ level of 284.3 ppm (SI Table 1). The MH experiment uses the same ice sheet and land-sea configurations as the PI but uses appropriate changes in orbital parameters and a $CO_2$ level of 264.4 ppm (SI Table 1). The MPI-ESM1.2-LR LGM experiment uses the ICE6G_C ice sheet and corresponding modification in land-sea geography, appropriate orbital parameters and a $CO_2$ level of 190 ppm (SI Table 1). The LGM simulation was re-started from a previous LGM simulation and then spun-up for 3850 years.

The MPI-ESM1.2-LR model has a spectral resolution of T63 (192 x 96 longitude/latitude). The climate and tree cover outputs necessary to run the EEO-based models were down-scaled to a resolution of 0.5° using spline interpolation. The daily data necessary to run the EEO-based models was obtained from monthly data, also using nearest neighbour and bilinear interpolation. Although many previous vegetation modelling studies have used climate anomalies from a baseline experiment (e.g. LGM minus PI), here we used model outputs directly – because although the anomaly approach is well-suited to adjust climate variables, it cannot be used to adjust simulated tree cover.

## 2.3. Stein-Alpert decomposition

Climate, light availability and atmospheric $CO_2$ concentration have independent effects on plant growth. To evaluate the unique effects of these different factors, and potential synergies between them, on the changes in GPP between the PI and the LGM and MH experiments, we used the Stein-Alpert decomposition method (Stein and Alpert, 1993), an approach that has been previously shown to be useful in evaluating the impacts of different factors on past vegetation changes (e.g. Martin-Calvo and Prentice, 2015; Sato et al., 2021). We used the pre-industrial simulation as the reference case (f0) and ran a series of factorial experiments in which specific factors were changed to their LGM or MH conditions as follows:

    Experiment f1: LGM (or MH) climate, PI $CO_2$ and PPFD
    Experiment f2: LGM (or MH) $CO_2$, PI climate and PPFD
    Experiment f3: LGM (or MH) PPFD, PI climate and $CO_2$
    Experiment f12: LGM (or MH) climate and $CO_2$, PI PPFD
    Experiment f13: LGM (or MH) climate and PPFD, PI $CO_2$

245    Experiment f23: LGM (or MH) $CO_2$ and PPFD, PI climate
246    Experiment f123: LGM (or MH) climate, $CO_2$ and PPFD
247
248 The impact of each factor or combination of factors was then calculated as:
249
250   $\langle f1 \rangle = f1 - f0$
251   $\langle f2 \rangle = f2 - f0$
252   $\langle f3 \rangle = f3 - f0$
253   $\langle f12 \rangle = f12 - (f1 + f2) + f0$
254   $\langle f13 \rangle = f13 - (f1 + f3) + f0$
255   $\langle f23 \rangle = f23 - (f2 + f3) + f0$
256   $\langle f123 \rangle = f123 - (f12 + f13 + f23) + (f1 + f2 + f3) - f0$
257
258 where the first three experiments represent the influence of the single changed factor, the
259 second three experiments represent synergies between pairs of factors, and the final experiment
260 represents the three-way synergy between all three factors.
261
262 The comparisons can only be made for the common land area between the PI and each
263 palaeoclimate experiment. The LGM factorial experiments therefore have a baseline GPP value
264 for the f0 experiment that does not include the areas exposed by lowered sea level, although
265 these are considered in the full LGM experiment. The full LGM and MH experiments include
266 changes to both air pressure and tree cover; these are not considered in the factorial experiments
267 because preliminary analyses indicated that the impact of these changes on simulated global
268 GPP was less than 0.2PgC yr$^{-1}$ and therefore negligible.
269
270 **3. Results**
271
272 Simulated global GPP at the LGM was 83.9 PgC yr$^{-1}$ (Table 1), considerably lower than the
273 simulated global value during the pre-industrial period (109.6 PgC yr$^{-1}$). The largest reductions
274 in GPP compared to the pre-industrial baseline were in the northern hemisphere extra-tropics
275 (Figure 2, Table 2), which experienced a more than 50% reduction in GPP. There was a more
276 modest decrease (13%) in the southern extra-tropics and only a small decrease in the tropics
277 (3%). Part of the reduction (10.5 PgC yr$^{-1}$) in global GPP reflects the loss of vegetation from
278 areas that were covered by ice at the LGM; this was only partially compensated by vegetation
279 growth on the continental shelves exposed by the reduced sea level (8.3 PgC yr$^{-1}$). Although
280 there was a reduction overall and across most of the world, some regions experienced a small
281 increase in productivity at the LGM compared to the PI (Figure 3). These are all in now-arid
282 regions and the increase therefore presumably reflects the fact that moisture constraints on
283 vegetation growth were reduced in the colder climate of the LGM.

**Table 1**: *Contribution to global changes in gross primary production (GPP) in the Last Glacial Maximum (LGM), the mid-Holocene (MH), and the pre-industrial (PI) experiments. The table gives the global total in each experiment, the GPP of land exposed by lowered sea level at the LGM, the GPP of land that was covered by ice sheets at the LGM and was exposed in the MH and PI experiments, and GPP for the land area in common between all three experiments.*

|  | Total non-glaciated land area | Land area covered by ice at LGM | Land area exposed by lowered sea level at LGM | Common land area between the experiments |
|---|---|---|---|---|

| | | | | |
|---|---|---|---|---|
| GPP LGM | 83.9 PgC yr$^{-1}$ | n/a | 8.3 PgC yr$^{-1}$ | 75.5 PgC yr$^{-1}$ |
| GPP MH | 110.3 PgC yr$^{-1}$ | 10.6 PgC yr$^{-1}$ | n/a | 99.6 PgC yr$^{-1}$ |
| GPP PI | 109.6 PgC yr$^{-1}$ | 10.5 PgC yr$^{-1}$ | n/a | 99.1 PgC yr$^{-1}$ |

**Table 2**: *Regional contributions to total annual gross primary production (GPP) in the tropics, the northern extra-tropics (NET) and the southern extra-tropics (SET) in the Last Glacial Maximum (LGM), the mid-Holocene (MH), and the pre-industrial (PI) experiments.*

| | LGM | MH | PI |
|---|---|---|---|
| Tropics (25°N-25S) | 56.4 PgC yr$^{-1}$ | 57.7 PgC yr$^{-1}$ | 58.3 PgC yr$^{-1}$ |
| NET (>25°N) | 21.4 PgC yr$^{-1}$ | 46.2 PgC yr$^{-1}$ | 44.3 PgC yr$^{-1}$ |
| SET (>25°S) | 6.0 Pg C yr$^{-1}$ | 6.4 PgC yr$^{-1}$ | 6.9 PgC yr$^{-1}$ |

Simulated GPP increased to 110.3 PgC yr$^{-1}$ in the MH compared to 83.9 PgC yr$^{-1}$ at the LGM. Part of this increase (10.6 PgC yr$^{-1}$) was a result of vegetation growth in areas that were covered by ice sheets during the LGM. However, there were notable increases in the non-glaciated high latitudes (northern Siberia and Beringia), in tropical regions, and in areas influenced by MH monsoon expansion (Sahel, south-east Asia, southern African savannas and the South American cerrado) (Figure 2). GPP increased in the common area between the LGM and MH experiments by ca 32% (Table 1), with the largest increase in the NET (Table 2). The transition from the MH to the PI resulted in a very small decrease in global GPP (Figure 3. Simulated GPP in the MH was slightly higher (4%) than in the PI experiment in the northern extra-topics, although still lower than in the PI in other regions (Table 2).

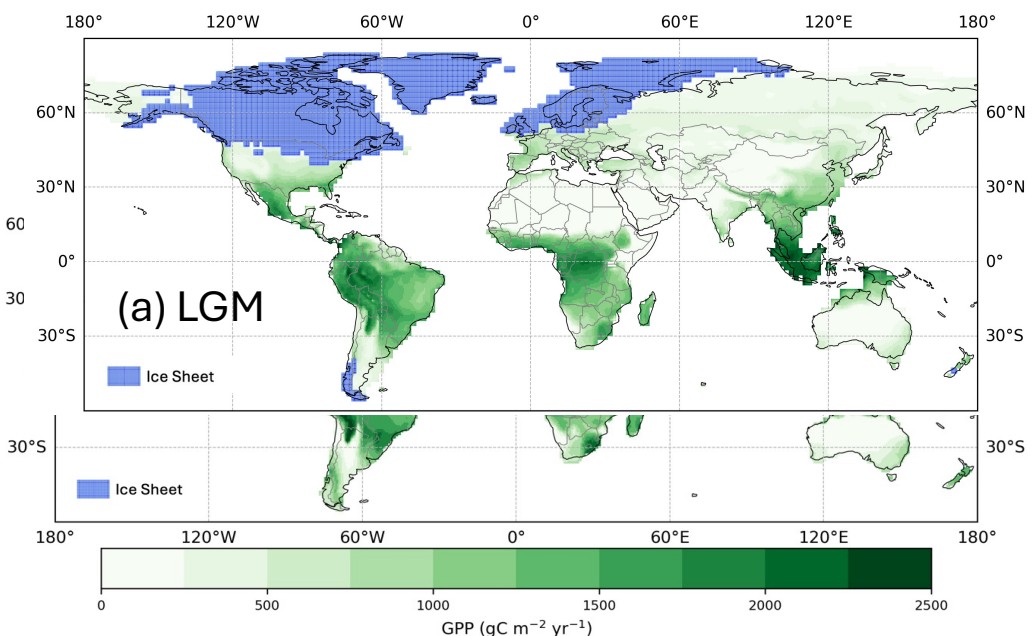

(a) LGM

(b) MH

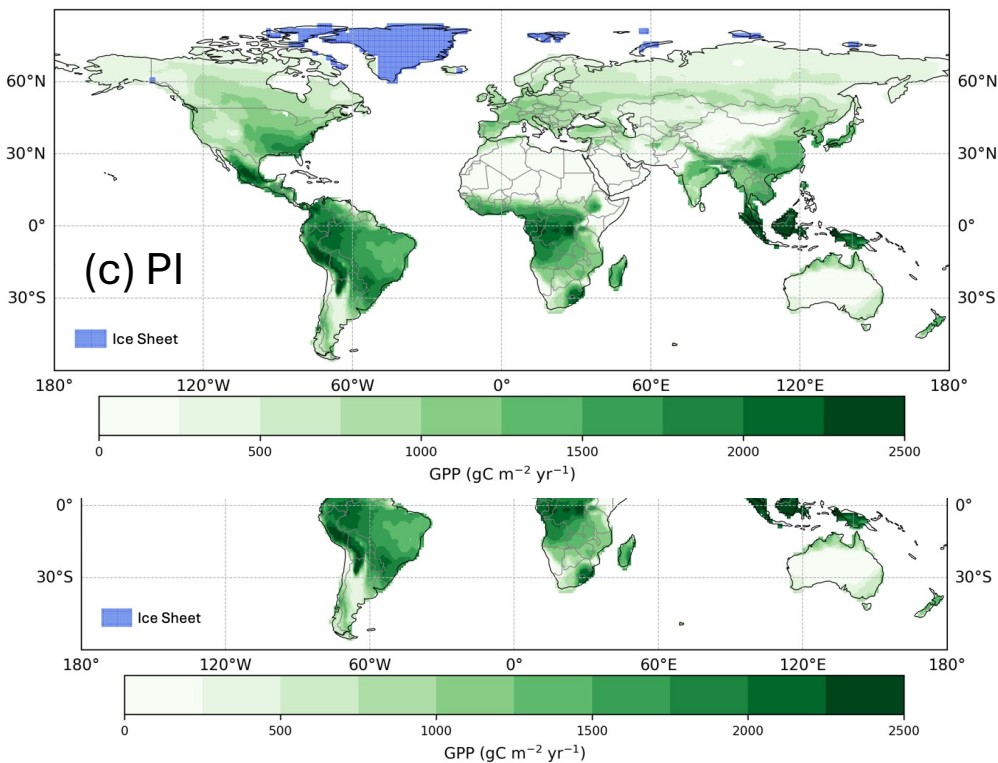

(c) PI

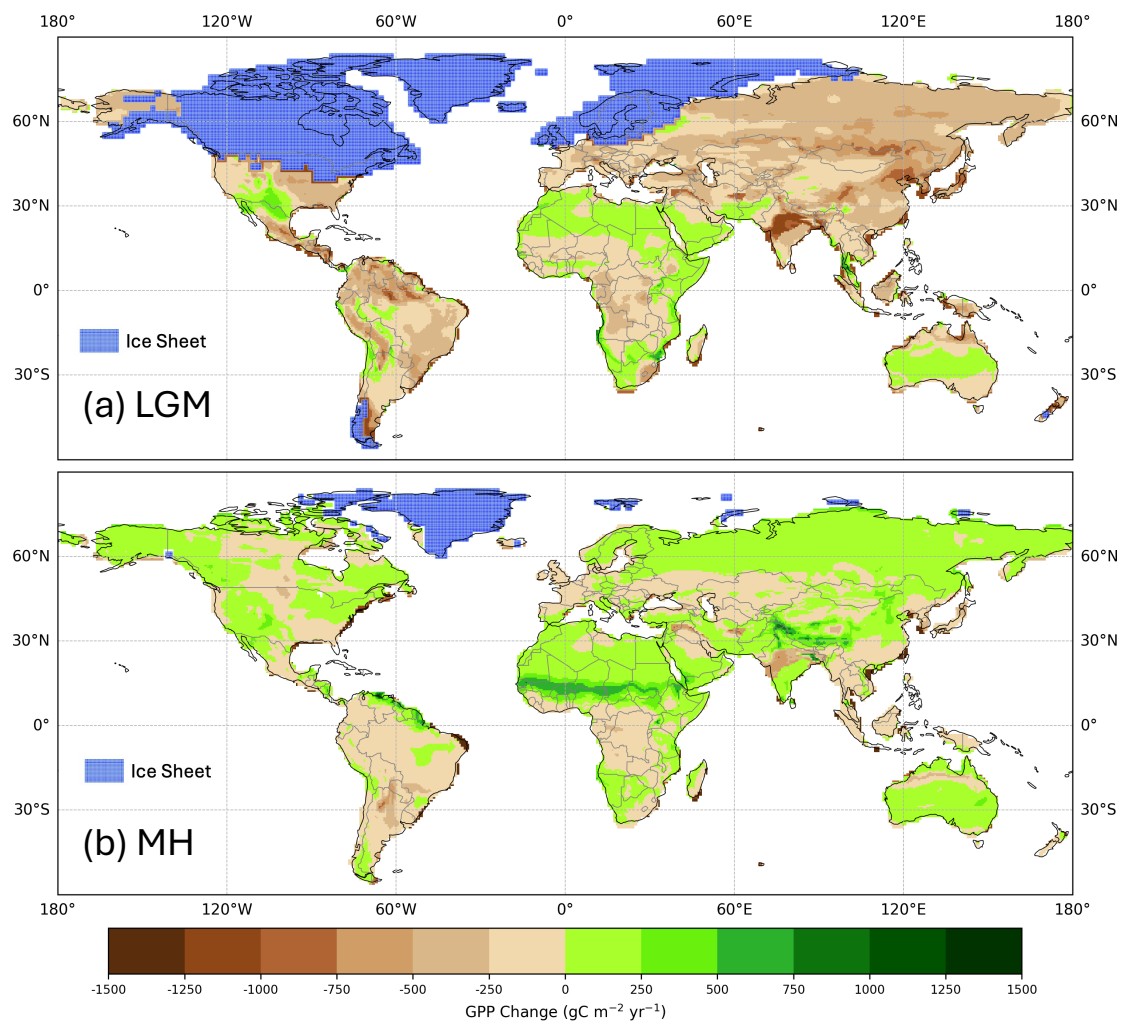

**Figure 3**: *Simulated change in total annual gross primary production (GPP) between the pre-*
*industrial (PI) and (a) the Last Glacial Maximum (LGM) and (b) the mid-Holocene (MH).*
These changes in GPP were accompanied by a shift in the relative importance of $C_3$ and $C_4$
plants (Table 3, Figure 4). $C_4$ plants represented 23% and 25% of the vegetation fraction in the
PI and MH experiments respectively, but 40% of the vegetation fraction at the LGM. $C_4$ plants
were responsible for 56% of the total GPP at the LGM compared to 25% and 21% in the MH
and PI respectively. The fraction of $C_4$ plants increased across most regions of the world at the
LGM (Supplementary Figure 1), but in some regions including the Central Great Plains of
North America, the northern Sahel, and the Tibetan Plateau and part of the Loess Plateau in
northeastern China $C_4$ plants were less abundant than in the PI. The areas where $C_4$ plants were
less abundant in the MH than in the PI were more extensive (Supplementary Figure 1) and are
primarily in regions of northern Africa and Asia influenced by the expansion of the monsoons.
**Table 3**: *Changes in $C_3$/$C_4$ fraction and contribution of $C_3$/$C_4$ vegetation to total GPP*

| | LGM | MH | PI |
|---|---|---|---|
| Global average $C_4$ fraction | 40% | 25% | 23% |
| Global average $C_3$ contribution of total annual GPP (gC m$^{-2}$ yr$^{-1}$) | 281.4 | 608.9 | 618.6 |
| Global $C_3$ contribution to total GPP (PgC **yr$^{-1}$**) | 37.1 | 82.8 | 86.2 |
| Global average $C_4$ contribution of total annual GPP (gC m$^{-2}$ yr$^{-1}$) | 297.7 | 166.3 | 140.5 |
| Global $C_4$ contribution to total GPP (PgC **yr$^{-1}$**) | 46.8 | 27.5 | 23.4 |

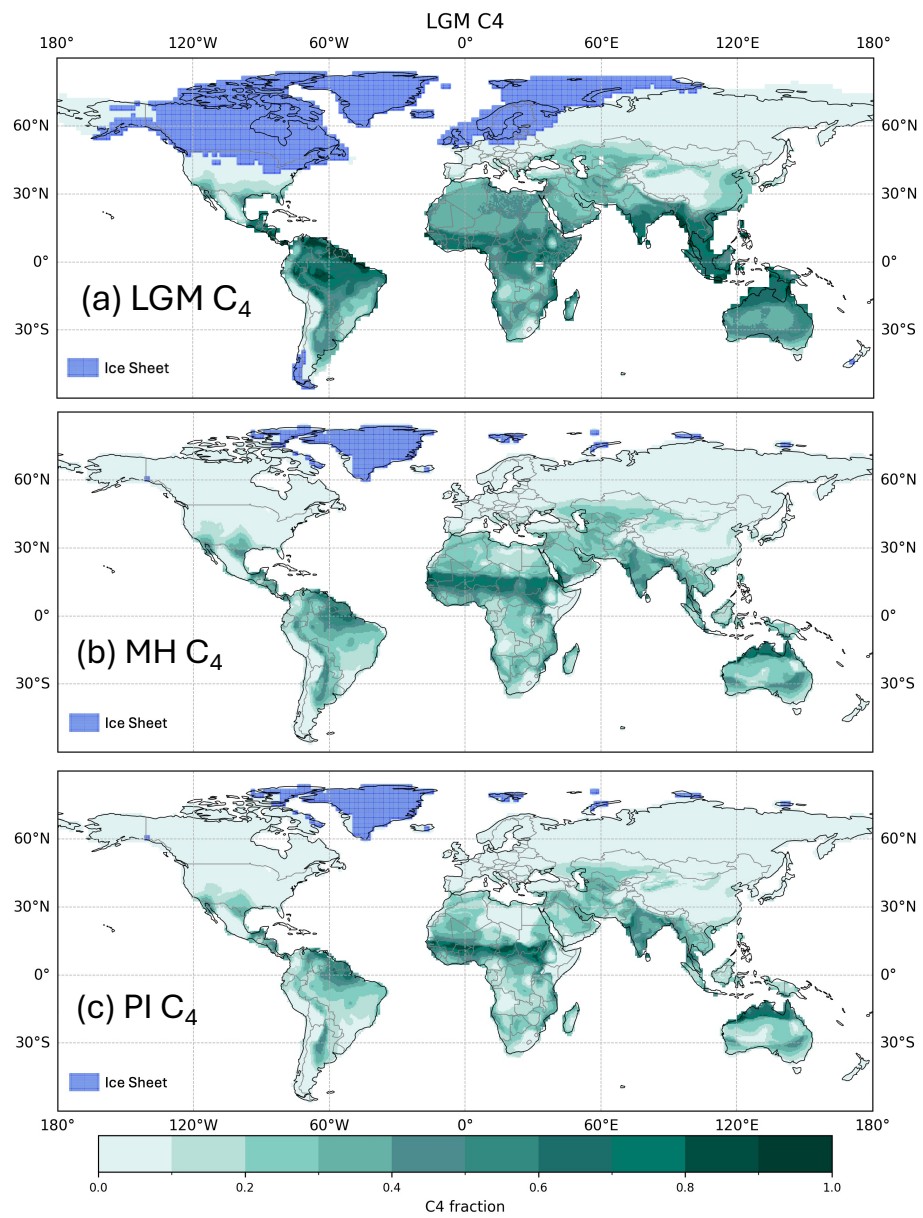

**Figure 4**. *Global C4 fraction distribution for (a) the Last Glacial Maximum (LGM), (b) the mid-Holocene (MH), and (c) the pre-industrial (PI).*

The factorial experiments showed that the changes in climate and $CO_2$ had a large negative effect on GPP at the LGM, while light (PPFD) had a small positive effect (Table 4, Figure 5). The shift to a colder, drier climate had a somewhat larger negative effect on plant productivity ($-14.8$ PgC yr$^{-1}$) than the reduction in $CO_2$ ($-12.2$ PgC yr$^{-1}$). Climate has a major impact on reducing GPP in the high- to mid-latitudes of North America and Eurasia (Figure 6a, Supplementary Figure 2) but changes due to the lowering of $CO_2$ were almost as important (Figure 6b, Supplementary Figure 3). Changes in climate (Supplementary Figure 2: Supplementary Table 2), most likely the overall reduction in precipitation (Supplementary Figure 5), was the most important factor causing reduced GPP in northern Amazonia, India

and north-western China. However, the cooler climate had a positive effect on GPP in regions that are semi-arid today (Supplementary Figure 2, Supplementary Figure 5). Changes in PPFD were the dominant factor in increasing GPP at the margin at the northernmost edge of the vegetated zone downwind of the Scandinavian ice sheet and into Beringia (Supplementary Figure 4).

The two-way synergy between climate and $CO_2$ was positive (Table 4, Figure 5), i.e. the change in GPP is less than would be expected if the impacts were additive. This reflects the fact that, whereas lower temperatures favour $C_3$ plants, lower $CO_2$ offsets this and promotes the expansion of $C_4$ plants over much of the globe (Supplementary Figures 6, 7). $C_4$ plants were especially favoured in tropical regions, where the climate changes were relatively muted, and the changes in $CO_2$ correspondingly more influential. The synergies of both climate and $CO_2$ with PPFD, although small (0.9 and 0.2 PgC yr$^{-1}$ respectively) are negative. The synergy between climate and PPFD probably reflects the fact reduced cloud cover in drier climates (Supplementary Figure 6, 8). The synergy between $CO_2$ and PPFD stems from the fact that both low $CO_2$ and high PPFD favour $C_4$ plants, increasing GPP particularly in the extratropics (Supplementary Figure 7, 8).

Climate changes had a positive effect on GPP in the mid-Holocene (Table 4, Figure 5). This likely reflects the impact of increased precipitation in now semi-arid regions due to monsoon expansion combined with warmer growing seasons in the high northern latitudes, both consequences of the orbitally-induced changes in solar radiation (Supplementary Figure 5). These experiments also show that changes in PPFD have a positive effect on plant growth, particularly in the northern mid- to high latitudes and in now-arid regions (Supplementary Figure 4). The positive impact in northern mid- to high latitudes appears to be due to enhancement of growing season conditions for $C_3$ plants, while the positive impact in now-arid regions reflects an increase in $C_4$ plants (Supplementary Figure 8). However, the reduction of $CO_2$ compared to the PI state (16 ppm) resulted in a much larger overall reduction in GPP than the enhancements due to climate or PPFD changes (Supplementary Figure 3). The impact of the lower $CO_2$ in the mid-Holocene is the dominant factor causing reductions in GPP in southern China, the southern hemisphere tropical and savanna regions in Africa, and in the cerrado of South America (Figure 6). The two-way synergies between the three drivers are all positive, but small (Table 4, Figure 5).

**Table 4**. *Stein-Alpert decomposition of the impact of changes in climate, $CO_2$ and light (photosynthetic photon flux density, PPFD), and their synergies, on gross primary production (GPP) at the Last Glacial Maximum (LGM) and in the mid-Holocene (MH) compared to the pre-industrial (PI) simulations. Note that the baseline GPP value for the LGM is for the common land area between this experiment and the PI simulation and is therefore smaller than the baseline GPP value for the MH decomposition.*

| Experiment | Stein-Alpert decomposition | Climate | $CO_2$ | PPFD | GPP (PgC yr$^{-1}$) |
|---|---|---|---|---|---|
| LGM | f0 | PI | PI | PI | 99.1 |
| | f1, LGM | LGM | PI | PI | 84.3 |
| | f2, LGM | PI | LGM | PI | 86.9 |
| | f3, LGM | PI | PI | LGM | 100.3 |
| | f12, LGM | LGM | LGM | PI | 75.4 |
| | f13, LGM | LGM | PI | LGM | 84.6 |
| | f23, LGM | PI | LGM | LGM | 87.8 |
| | f123, LGM | LGM | LGM | LGM | 75.7 |
| MH | f0 | PI | PI | PI | 109.6 |
| | f1, MH | MH | PI | PI | 111.5 |
| | f2, MH | PI | MH | PI | 107.0 |
| | f3, MH | PI | PI | MH | 110.6 |
| | f12, MH | MH | MH | PI | 109.1 |
| | f13, MH | MH | PI | MH | 112.5 |
| | f23, MH | PI | MH | MH | 108.1 |
| | f123, MH | MH | MH | MH | 110.1 |

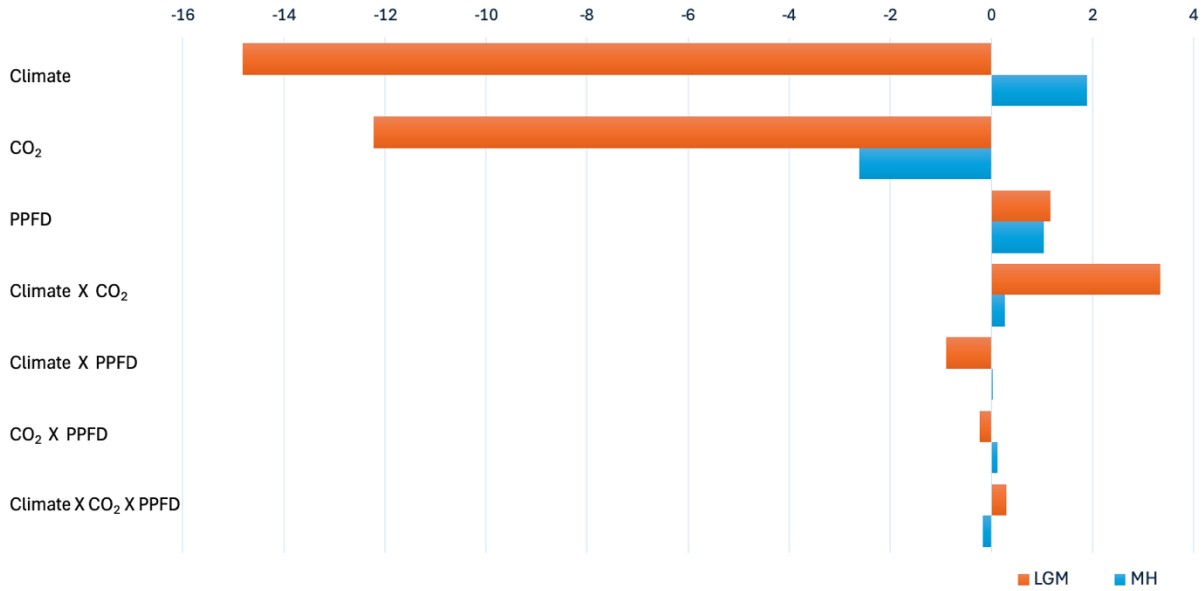

*Figure 5. Impact of climate, light and $CO_2$ on the changes in gross primary production (GPP, PgC) at the Last Glacial Maximum (LGM) and the mid-Holocene (MH) compared to the pre-industrial (PI) period. Note that the results are based on the common land area between each experiment and the PI simulation.*


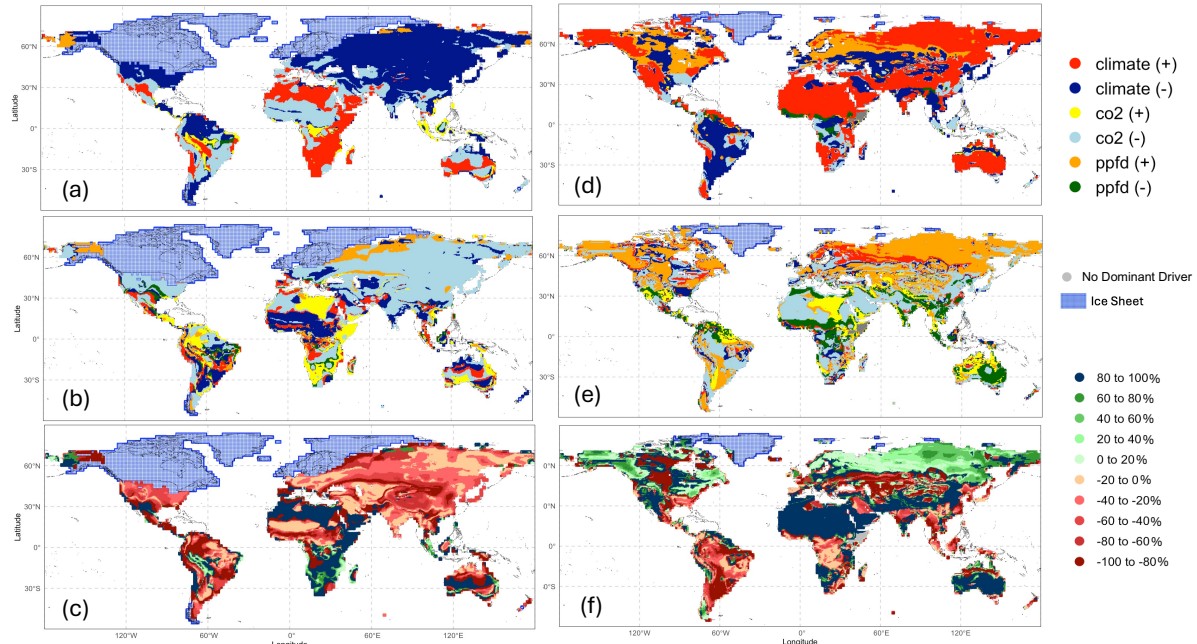

*Figure 6. Global distribution of (a) main drivers and constraints (b) secondary drivers and*
*constraints and (c) the proportional difference (percentage) of total change between the main*
*and the secondary driver on gross primary production (GPP) at the Last Glacial Maximum*
*(LGM) compared to the pre-industrial (PI) experiment; and (d) main drivers and constraints*
*(e) secondary drivers and constraints (f) the proportional difference (percentage) of total*
*change between the main and the secondary driver on gross primary production (GPP) in the*
*Mid-Holocene (MH) compared to the pre-industrial (PI) experiment.*

**4. Discussion**
We have shown that the LGM was characterised by a large reduction in modelled GPP, while
the mid-Holocene was characterised by a small increase in GPP compared to the pre-industrial
state. Estimated GPP at the LGM was ca 84 PgC yr$^{-1}$ compared to ca 110 PgC yr$^{-1}$ in the PI.
The simulated reduction at the LGM is consistent with previous model-based estimates (e.g.
Francois et al., 1998; Prentice et al., 2011; Hoogakker et al., 2016), including those from the
latest phase of the Couple Model Intercomparison project (CMIP6/PMIP4: Supplementary
Table 3). However, previous estimates of GPP span a considerable range, from 40-110 PgC yr$^{-1}$
$^{1}$). This reflects differences in the boundary conditions used, differences in the vegetation
models used and their sensitivity to changes in CO2, and differences in the structure and
parameterisations of the climate models overall. Diagnosing the specific causes of this large
range is therefore extremely difficult. The parameter sparse nature of our EEO-based modelling
approach, and the fact that the individual processes that give rise to the simulated GPP have
been independently validated, suggest that our estimate of ca 84 PgC yr$^{-1}$ is more likely to be
realistic than previous estimates. A limited number of studies have estimated GPP at the LGM
by constraining model estimates using oxygen isotope records from ice cores (Landais et al.,
2007; Ciais et al., 2011; Yang et al., 2022). The still large range in simulated GPP (40-110
PgC yr$^{-1}$) reflects, in part, uncertainties associated with estimating ocean productivity and
respiration fractionation rates. Thus, although there is a consensus that GPP was considerably
lower at the LGM than during pre-industrial times, and this is consistent with pollen evidence
for a very large reduction in tree cover over much of the world (Prentice et al., 2000; Williams,
2003; Pickett et al., 2004; Marchant et al., 2009), the absolute magnitude of this change is

uncertain. Nevertheless, since the climate simulated by the MPI ESM has been shown to reproduce pollen-based climate reconstructions better than most other CMIP6/PMIP4 models (Kageyama et al., 2021) and we use robust EEO-models to estimate the change in GPP, the partitioning of the impacts of different factors in the simulated reduction of GPP is likely to be robust.

The modelled abundance of $C_4$ plants was nearly double at the LGM compared to the pre-industrial era (40% versus 23% of the vegetation fraction) and that $C_4$ vegetation was responsible for 56% of the total modelled GPP at that time. These changes are broadly consistent with pollen-based reconstructions, indicating a substantial reduction in tree cover at the LGM (Prentice et al., 2000). It is difficult to estimate the magnitude of this reduction because existing regional reconstructions have not been applied to the LGM (e.g. Zanon et al., 2018; Serge et al., 2023) and furthermore employ techniques that are based on modern calibrations and therefore do not account for the impact of $CO_2$ on tree cover (Prentice et al., 2022). However, while pollen data can be used discriminate between trees (virtually all $C_3$) and grasses, it cannot be used to infer changes in the importance of $C_3$ and $C_4$ grasses. Compound-specific $\delta^{13}C$ analyses of leaf wax biomarkers provide evidence of the relative contribution of $C_3$ and $C_4$ plants (Eglinton & Eglinton, 2008; Diefendorf et al., 2010) and have shown that $C_4$ plants were more abundant at the LGM than during the Holocene in many regions of the world (e.g. in southern Africa: Rommerskirchen et al., 2006; Vogts et al., 2012; eastern Africa: Sinninghe Damsté et al., 2011; Himalayan Basin: Galy et al., 2008; southern China: Jiang et al., 2019; south-western North America: Cotton et al., 2016; northern South America: Makou et al., 2007), consistent with our simulations. There are a few regions where $C_4$ plants were less abundant at the LGM than during the Holocene, including the Chinese Loess Plateau and the Great Plains of North America (Cotton et al., 2016). Both of these regions are identified as characterised by reduced $C_4$ abundance in our simulations. The consistency of the signs of the regional changes in the observed relative abundance of $C_3$ to $C_4$ plants to our simulated changes provides strong support for the model predictions. A number of modelling studies have shown that $C_4$ plants were globally more abundant at the LGM (e.g. Harrison & Prentice, 2003; Bragg et al., 2013; Martin Calvo & Prentice, 2015) but did not quantify the relative contribution of $C_4$ plants to global GPP. Thus, our analyses are consistent with previous studies of the nature of the shift in vegetation composition at the LGM and provide, for the first time, a quantitative estimate of the magnitude of this change.

Climate has a negative effect on GPP at the LGM but a positive effect in the MH. The LGM climate was globally colder and drier, although the largest changes in both temperature and precipitation were in the northern mid- to high-latitudes (Kageyama et al., 2021). This is reflected in our simulations; the overall reduction in GPP compared to the pre-industrial baseline in the northern extra-tropics was 52%, far larger than the reductions in the southern extra-tropics (13%) or the tropics (3%). The cooling in the ice-free regions of the northern extra-tropics reflects advection of cold air temperatures downwind from the ice sheets, while the drying largely reflects the temperature-induced reduction in evaporation and precipitation recycling (Izumi et al., 2013; Li et al., 2013; Kageyama et al., 2021). The positive effect of climate on GPP in the MH reflects changes in precipitation in now semi-arid regions of the sub-tropics as a result of the expansion of the northern hemisphere monsoons and a lengthening of the growing season in the northern mid- to high-latitudes as a result of increased solar radiation in summer (Brierley et al., 2020). These changes in climate are reflected in our simulations; although the northern extra-tropics are the only region to show an overall increase in GPP compared to the pre-industrial (4%), regions influenced by monsoon expansion, such as the Sahel and parts of Asia, also show increased GPP.

The modelled reduction of GPP by low LGM relative to pre-industrial $CO_2$ was of similar magnitude (12%) to that of LGM climate (15%). Some other factorial model experiments (e.g. O'Ishi and Abe-Ouchi, 2013; Claussen et al., 2013; Martin Calvo and Prentice, 2015; Chen et al., 2019; Haas et al., 2023; see Supplementary Table 4) have shown a larger impact of $CO_2$ on primary production (either GPP or net primary production, NPP) relative to climate. For example, Claussen et al. (2013) showed reductions in NPP of 4% due to climate and 45% due to $CO_2$ and Martin Calvo and Prentice (2015) showed reductions in NPP of 2% due to climate and 23% due to $CO_2$. Some of differences among experiments may have been caused by difference in modelled climate (Haas et al., 2023); but changes in PFT abundance are likely to be an important additional source of uncertainty. Woillez et al. (2011) also indicate a dominant role for low glacial $CO_2$ in reducing NPP at the LGM. In that analysis, however, a greater sensitivity of needleleaf PFTs to low $CO_2$ compared to brodleaf PFTs was implied by choices of parameter values that were not necessarily well-founded, and led to an unrealistically large simulated extent of broad-leaved forests at the LGM.

In addition to the fact that these various experiments were based on different models of the LGM climate, they were also made using different biosphere models (Supplementary Table 4) – which may have different sensitivities to $CO_2$ changes. Thus, although models agree that changes in $CO_2$ contributed to the large observed differences between LGM and pre-industrial vegetation patterns, the magnitude of the impact of low $CO_2$ on primary production is still uncertain. The modelled impact of lowered $CO_2$ on GPP in the MH here is larger than the impact of climate, offsetting the positive impacts of climate change in the MH experiment. The importance of $CO_2$ in driving vegetation changes has been widely commented on for the LGM (Polley et al., 1993; Jolly & Haxeltine,1997; Cowling & Sykes, 1999; Harrison & Prentice, 2003; Flores et al., 2009; Prentice et al., 2011; Bragg et al., 2013; Martin Calvo & Prentice, 2015) and in the context of ongoing and future climate changes (Piao et al., 2006; Keenan et al., 2014; Archer et al., 2017; Haverd et al., 2020: Piao et al., 2020) but its role in offsetting the positive impacts of climate change in the MH has not been widely noted. The simulated overall change in GPP in the MH compared to the PI is small (< 1 PgC yr$^{-1}$). Nevertheless, the changes in response to individual drivers are consistent with expectations: changes in climate and PPFD had a positive impact on GPP while the reduction in $CO_2$ in the MH compared to the PI had a negative impact on GPP. The positive effect of climate on GPP in the MH reflects changes in precipitation in now semi-arid regions of the sub-tropics, as a result of the orbitally induced expansion of the northern hemisphere monsoons and the lengthening of the growing season in the northern mid- to high-latitudes (Brierley et al., 2020). These changes in climate are reflected in our simulations. The northern extratropics are the only region to show an overall increase in GPP compared to the pre-industrial (4%) when $CO_2$ effects are included, but regions influenced by monsoon expansion, such as the Sahel and parts of South and East Asia, also show a tendency to increased GPP due to the MH climate.

We have derived climate inputs from the MPI ESM. When compared to reconstructions of both marine and terrestrial climate variables, the MPI ESM has been shown to be among the best-performing models both for the LGM and the mid-Holocene (Brierley et al., 2020; Kageyama et al., 2021). Nevertheless, the use of a single climate model is a limitation of this study. It would be useful to repeat these analyses with a wider range of models that have made palaeoclimate simulations of these two key periods, but the constraint is that most of these models do not provide information on changes in tree cover that is to run the $C_3/C_4$ competition model.

We have used a sequence of EEO-based models to simulate GPP and the relative contribution of $C_3$ and $C_4$ plants to overall productivity. Haas et al. (2023) also used the P model to simulate GPP at the LGM. Other studies of past vegetation changes have used models that simulate changes in past vegetation on the basis of the competition between PFTs. PFT-based models require key physiological parameters to be specified separately for each PFT. The EEO modelling approaches used here avoid this complexity, considerably reducing uncertainties due to model parameterisation (Harrison et al, 2021) while at the same time representing the key processes of photosynthesis and plant growth accurately (Wang et al., 2017; Smith et al., 2019; Jiang et al., 2020; Lavergne et al., 2020; Peng et al., 2020; Smith & Keenan, 2020; Wang et al., 2020; Xu et al., 2021; Zhu et al., 2022). Furthermore, they capture recent trends in vegetation growth more accurately than the land-surface models used to predict the terrestrial carbon cycle (Cai et al., 2025; Zhou et al., 2025). Given their simplicity, the fact that the very few parameters required are well constrained from observations, and the demonstrated quality of their performance, EEO-modelling holds considerable promise for understanding past vegetation changes and their impact on climate.

## 5. Conclusions

Eco-evolutionary optimality approaches provide a robust way of modelling vegetation changes under different climate regimes. We compared simulated changes in GPP and $C_3/C_4$ plant abundance in a cold glacial and a warm interglacial period relative to the pre-industrial state. We showed that the colder, drier climate at the LGM substantially decreases GPP and the warmer, wetter climate of the MH increases GPP. Changes in vegetation productivity caused by the lower $CO_2$ in both intervals compared to the pre-industrial contributed to the reduction of GPP at the LGM and was sufficient to annul the positive impacts of climate on GPP during the MH. These results point to the importance of a realistic treatment of the direct physiological impacts of $CO_2$ on plant growth to simulate realistic ecosystem changes, both in the past and in the future.

**Data Availability**
The CMIP6 MPI-ESM1-2-LR outputs are accessible via the Earth System Grid Federation (ESGF) at http://esgf-node.llnl.gov/search/cmip6/ (last accessed: 2 December 2024). Interpolated input data and derived outputs related to this study are available at DOI: 10.5281/zenodo.14257604. The documentation for the P model, the C3/C4 competition model, and the SPLASH model can be found at DOI: 10.5281/zenodo.8366848 (Orme and Marion, 2023). The codes used for model coupling and experiment analysis used in this paper is available at DOI: 10.5281/zenodo.14257604.

**Supplement.**
Supplementary Information is available for this paper.

**Author Contributions**
JZ, SPH and ICP designed the study. BZ provided model code. JZ ran the experiments. JZ and SPH conducted the analyses. SPH wrote the first version of the manuscript and all co-authors contributed to the final version.

**Competing Interests**
None of the authors has any competing interests.

**Financial Support and Acknowledgements**

JZ and SPH acknowledge NERC funding for the project "When and Why does it Rain in the
Desert: Utilising unique stalagmite and dust records on the northern edge of the Sahara". This
work is a contribution to the LEMONTREE (Land Ecosystem Models based On New Theory,
obseRvations and ExperimEnts) project (SPH, ICP). LEMONTREE research received support
through Schmidt Sciences, LLC. ICP also acknowledges funding from the European Research
Council for the project REALM (Re-inventing Ecosystem And Land-surface Models, Grant
Number 787203).

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
