# Peer review of "Eco-evolutionary Modelling of Global Vegetation Dynamics and the Impact of CO2"

_EGUsphere, 2024_

## Author Comment (AC1)

We thank the reviewer for their comments on the paper. Our response to the specific points is given below in *italics* and proposed changes to the text in blue.

1) However, I have concerns regarding the novelty and broader implications of this work, particularly how it advances beyond previous factorial simulations.

*A number of published studies have examined the modelled global impacts of climatic versus physiological $CO_2$ effects on LGM vegetation. Several have simulated LGM climate impacts on vegetation (and/or fire) with, or without, inclusion of the physiological effects of $CO_2$ on plants (Levis et al. 1999, Harrison and Prentice 2003, Martin Calvo et al. 2014). Others have performed factorial experiments to more formally separate the effects of climate and $CO_2$ (Woillez et al. 2011, O'ishi & Abe-Ouchi 2013, Claussen et al. 2013, Martin Calvo & Prentice 2015, Chen et al. 2019, Haas et al. 2023). The strong influence of low LGM $CO_2$ (in addition to effects of a cold and dry global climate) in suppressing primary production is a consistent finding from both types of analysis. However, there are substantial differences in the relative importance of $CO_2$ and climate change effects attributed by different models.*

*All these previous studies have used land ecosystem models based on the "plant functional type" (PFT) paradigm, which requires lists of parameter values to be specified separately for each PFT. This approach embeds uncertainty in the delimitation of PFTs and the parameter values assigned to them, because in reality trait variation within PFTs is substantially larger than variation between them (Kattge et al., 2011). In some cases, the PFT representation has resulted in an unrealistic simulation of LGM vegetation patterns (e.g. Woillez et al. 2011). Our approach here is intended to put the calculation of changes in GPP on a sounder basis by using a recently developed model, the P model, which is expressly designed to account for acclimation and adaptation to environment **independently** of PFTs (apart from the distinction between the $C_3$ and $C_4$ photosynthetic pathways). The P model has been subject to extensive evaluation against worldwide data from eddy-covariance flux towers. We use a newly published extension of the model which simulates foliage cover and its seasonal cycle, also independently of PFTs, and has been shown to do so more accurately than any state-of-the-art vegetation model (Zhou et al., 2025). We use a new process-based scheme to represent the relative competitive success of $C_3$ versus $C_4$ plants, which has been validated against worldwide soil carbon stable isotope data (Lavergne et al., 2024). This combination of parameter-sparse and independently validated models enables us, for the first time, to apply an eco-evolutionary optimality approach to simulate past vegetation function in a globally consistent way.*

*Less attention has been paid to productivity changes since the MH, compared to the LGM. As the $CO_2$ and climate differences between MH and pre-industrial time are relatively small, it is likely that primary production changes as represented by conventional models would be dominated by uncertainties linked to PFTs. Here, we use the same consistent global methodology to estimate MH to pre-industrial changes in GPP. We include the effect of changes in the light regime, which are a consequence of changes in the seasonal and latitudinal distribution of insolation due to orbital forcing, as well as changes in cloud cover linked to monsoon shifts.*

*In order to make it clear what the novel contribution of this paper is relative to earlier work, we propose to modify the Introduction as follows (line 62 et seq.):*

Three sets of factors could potentially impact vegetation productivity changes between the LGM, MH and pre-industrial periods: changes in climate, atmospheric $CO_2$ and solar radiation. Several published studies have simulated LGM climate impacts on vegetation (and/or fire, interacting with vegetation), with – or without – consideration of the additional physiological effects of low $CO_2$ on plants (Levis et al. 1999, Harrison and Prentice 2003, Martin Calvo et al. 2014). Other studies have performed factorial experiments to more formally separate the effects of climate and $CO_2$ (Woillez et al. 2011, O'ishi & Abe-Ouchi 2013, Claussen et al. 2013, Martin Calvo & Prentice 2015, Chen et al. 2019, Haas et al. 2023).

Comparison among these studies of LGM-to-recent primary production shifts is approximate at best because they have used different climate models and experimental protocols. Some have used pre-industrial conditions as a reference; others, modern (higher-$CO_2$) conditions. However, they all have used land ecosystem models based on the plant functional type (PFT) concept. Uncertainty in the delimitation of PFTs and the parameter values assigned to them is endemic to this type of model, as variation of quantitative traits within PFTs in the real world is generally much larger than variation between them (Kattge et al., 2011). In some cases, the model PFT representation has resulted in an unrealistic simulation of LGM vegetation patterns (e.g. Woillez et al. 2011). Here we use the P model (Stocker et al. 2020), which accounts for acclimation and adaptation to environment independently of PFTs on the basis of universal eco-evolutionary optimality (EEO) hypotheses. The P model has been subject to extensive evaluation against worldwide data from eddy covariance flux towers across all biomes. We include an extension of the P model which simulates foliage cover and its seasonal cycle – also independently of PFTs. This extended model has been shown to reproduce foliage amounts and seasonal dynamics more accurately than any state-of-the-art vegetation model (Zhou et al., 2025). We use a simple process-based scheme to represent the relative competitive success of $C_3$ versus $C_4$ plants, which has been validated against worldwide soil carbon stable isotope data (Lavergne et al., 2024). This combination of three independently tested, PFT-independent modelling components enables us, for the first time, to apply an EEO-based approach to simulate LGM and recent vegetation function in a globally uniform way

There has been some work on the implications of MH climate for biome distributions (e.g. Kaplan et al., 2003; Wohlfahrt et al., 2008) but little consideration of the impacts of climate and $CO_2$ on global productivity changes since the MH (Foley, 1994; François et al. 1999). Here, we use the same consistent methodology that we apply to the LGM to estimate MH-to-pre-industrial changes in global GPP. Our analysis includes the effect of changes in the light regime, which are a necessary consequence of changes in the seasonal and latitudinal distribution of insolation due to orbital forcing, as well as changes in cloud cover linked to monsoon shifts.

EEO-based modelling approaches provide parameter-sparse representations of plant and vegetation processes, thus considerably reducing uncertainties due to model parameterisation (Harrison et al, 2021). They have been shown to perform as well or better than more complex models under recent conditions (Cai et al., 2025; Zhou et al., 2025) and thus can provide a robust way of modelling vegetation changes under different climate regimes. We use a series of counter-factual experiments ….

*We will remove Bragg et al. (2013) from this section since that was not a global study, although it is referred to later on when we discuss regional patterns. We will add the following references:*

Chen, W., Zhu, D., Ciais, P., Huang, C., Viovy, N., Kageyama, M., 2019. Response of vegetation cover to $CO_2$ and climate changes between Last Glacial Maximum and pre-industrial period in a dynamic global vegetation model. Quaternary Science Reviews 218, 293-305, https://doi.org/10.1016/j.quascirev.2019.06.003

Claussen, M., Selent, K., Brovkin, V., Raddatz, T. Gayler, V., 2013. Impact of CO2 and climate on Last Glacial maximum vegetation – a factor separation. Biogeosciences 10, 3593-360. https://bg.copernicus.org/articles/10/3593/2013/

Foley, J. A. (1994), The sensitivity of the terrestrial biosphere to climatic change: A simulation of the Middle Holocene, *Global Biogeochem. Cycles*, 8(4), 505–525, doi:10.1029/94GB01636.

Francois, L., Godderis, Y., Warnant, P., Ramstein, G., de Noblet, N., and Lorenz, S.: Carbon ´ 5 stocks and isotopic budgets of the terrestrial biosphere at mid-Holocene and last glacial maximum times, Chem. Geol., 159, 163–189, 1999. 972

Haas, O., Prentice, I.C., Harrison, S.P., 2023. Examining the response of wildfire properties to climate and atmospheric $CO_2$ change at the Last Glacial Maximum *Biogeosciences* 20: 3981-3995, https://doi.org/10.5194/bg-20-3981-2023

Harrison, S.P., Cramer, W., Franklin, O., Prentice, I.C., Wang, H., Brännström, Å., de Boer, H., Dieckmann, U., Joshi, J., Keenan, T.F., Lavergne, A., Manzoni, S., Mengoli, G., Morfopoulos, C., Peñuelas, J., Pietsch, S., Rebel, K.T., Ryu, Y., Smith, N.G., Stocker, B.D., Wright, I.J., 2021. Eco-evolutionary optimality as a means to improve vegetation and land-surface models. *New Phytologist* 231: 2125-2141, https://doi.org/10.1111/nph.17558

Kattge, J., Díaz, S., Lavorel, S., Prentice, I.C., Leadley, P., Bönisch, G., Garnier, E., Westoby, M., Reich, P.B., Wright, I.J., Cornelissen, J.H.C., Violle, C., **Harrison, S.P.**, van Bodegom, P.M., Reichstein, M., Soudzilovskaia, N.A., Ackerly, D.D., Anand, M., Atkin, O., Bahn, M., Baker, T.R., Baldocchi, D., Bekker, R., Blanco, C., Blonder, B., Bond, W., Bradstock, R., Bunker, D.E., Casanoves, F., Cavender-Bares, J., Chambers, J., Chapin, F.S., Chave, J., Coomes, D., Cornwell, W.K., Craine, J.M., Dobrin, B.H., Durka, W., Elser, J., Enquist, B.J., Esser, G., Estiarte, M., Fagan, W.F., Fang, J., Fernández, F., Fidelis, A., Finegan, B., Flores, O., Ford, H., Frank, D., Freschet, G.T., Fyllas, N.M., Gallagher, R., Green, W., Gutierrez, A.G., Hickler, T., Higgins, S., Hodgson, J.G., Jalili, A., Jansen, S., Kerkhoff, A.J., Kirkup, D., Kitajima, K., Kleyer, M., Klotz, S., Knops, J.M.H., Kramer, K., Kühn, I., Kurokawa, H., Laughlin, D., Lee, T.D., Leishman, M., Lens, F., Lenz, T., Lewis, S.L., Lloyd, J., Llusià, J., Louault, F., Ma, S., Mahecha, M.D., Manning, P., Massad, T., Medlyn, B., Messier, J., Moles, A., Müller, S., Nadrowski, K., Naeem, S., Niinemets, U., Nöllert, S., Nüske, A., Ogaya, R., Joleksyn, J., Onipchenko, V.G., Onoda, Y., Ordoñez, J., Overbeck, G., Ozinga, W., Patiño, S., Paula, S., Pausas, J.G., Peñuelas, J., Phillips, O.L., Pillar, V., Poorter, H., Poorter, L., Poschlod, P., Proulx, R., Rammig, A., Reinsch, S., Reu, B., Sack, L., Salgado, B., Sardans, J., Shiodera, S., Shipley, B., Sosinski, E., Soussana, J-F., Swaine, E., Swenson, N., Thompson, K., Thornton, P., Waldram, M., Weiher, E., White, M., Wright, S.J., Zaehle, S., Zanne, A.E., Wirth, C., 2011. TRY – a global database of plant traits. *Global Change Biology* 17: 2905–2935. doi:10.1111/j.1365-2486.2011.02451.x

Martin Calvo, M., Prentice, I.C., Harrison, S.P., 2014. Climate versus carbon dioxide controls on biomass burning: a model analysis of the glacial-interglacial contrast. *Biogeosciences*, 11, 6017–6027. doi:10.5194/bg-11-6017-2014

O'ishi, R. and Abe-Ouchi, A.: Influence of dynamic vegetation on climate change and terrestrial carbon storage in the Last Glacial Maximum, Clim. Past, 9, 1571–1587, https://doi.org/10.5194/cp-9-1571-2013, 2013

Levis S, Foley JA, Pollard D. CO$_2$, climate, and vegetation feedbacks at the Last Glacial Maximum. *J Geophys Res* 1999, **104**: 31191–31198.

We will also take the opportunity to update two references which are now published:
Cai, W., Zhu, Z., Harrison, S.P., Ryu, Y., Wang, H., Zhou, B., Prentice, I.C., 2025. A unifying principle for global greenness patterns and trends *Nature Communication and Environment* 6, 19, https://doi.org/10.1038/s43247-025-01992-0
Zhou, B., Cai, W., Zhu, Z., Wang, H., Harrison, S.P., Prentice, I.C., 2025. A general model for the seasonal to decadal dynamics of leaf area *Global Change Biology* e70125, https://doi.org/10.1111/gcb.70125

*Previous studies used a variety of climate inputs and vegetation models. We will add a table in Supplementary (Supplementary Table 5) summarising these experiments and add text with a more detailed comparison between the various experiments in the Discussion, as follows:*

The modelled reduction of GPP by low LGM relative to pre-industrial CO$_2$ was of similar magnitude (12%) to that of LGM climate (15%). Some other factorial model experiments (e.g. O'Ishi and Abe-Ouchi, 2013; Claussen et al., 2013; Martin Calvo and Prentice, 2015; Chen et al., 2019; Haas et al., 2023; see Supplementary Table 5) have shown a larger impact of CO$_2$ on primary production (either GPP or net primary production, NPP) relative to climate. For example, Claussen et al. (2013) showed reductions in NPP of 4% due to climate and 45% due to CO$_2$ and Martin Calvo and Prentice (2015) showed reductions in NPP of 2% due to climate and 23% due to CO$_2$. Some of differences among experiments may have been caused by difference in modelled climate (Haas et al., 2023); but changes in PFT abundance are likely to be an important additional source of uncertainty. Woillez et al. (2011) also indicate a dominant role for low glacial CO$_2$ in reducing NPP at the LGM. In that analysis, however, a greater sensitivity of needleleaf PFTs to low CO$_2$ compared to brodleaf PFTs was implied by choices of parameter values that were not necessarily well-founded, and led to an unrealistically large simulated extent of broad-leaved forests at the LGM.

In addition to the fact that these various experiments were based on different models of the LGM climate, they were also made using different biosphere models (Supplementary Table 5) – which may have different sensitivities to CO$_2$ changes. Thus, although models agree that changes in CO$_2$ contributed to the large observed differences between LGM and pre-industrial vegetation patterns, the magnitude of the impact of low CO$_2$ on primary production is still uncertain. The modelled impact of lowered CO$_2$ on GPP in the MH here is larger than the impact of climate, offsetting the positive impacts of climate change in the MH experiment. The importance of CO$_2$ in driving vegetation changes has been widely commented on for the LGM (Polley et al., 1993; Jolly & Haxeltine,1997; Cowling & Sykes, 1999; Harrison & Prentice, 2003; Flores et al., 2009; Prentice et al., 2011; Bragg et al., 2013; Martin Calvo & Prentice, 2015) and in the context of ongoing and future climate changes (Piao et al., 2006; Keenan et al., 2014; Archer et al., 2017; Haverd et al., 2020: Piao et al., 2020) but its role in offsetting the positive impacts of climate change in the MH has not been widely noted. Despite the small change in CO$_2$ between the PI and MH (16 ppm), according to our simulations the lowering of CO$_2$ would have reduced GPP by about 3 PgC whereas the increase produced by the change in climate is only 2 PgC.

2) The impacts of changing CO$_2$ levels on plant growth have already been incorporated into many land surface models (LSMs) and Earth System Models (ESMs), though the

magnitude of physiological effects differs between models. The authors should clarify how this study advances prior work and explicitly discuss its implications for ecosystem modeling.

*The impact of changing CO₂ levels on plant and ecosystem function is represented in most LSMs but its magnitude varies considerably among models, indicating uncertainty about how it should be implemented. The large number of PFT-specific parameters that need to be specified in state-of-the-art LSMs further increases uncertainty in model predictions of the response to CO₂. However, we have confidence in the response of the P model to changing CO₂ because (a) it arises naturally from the model's foundation in the biochemistry of photosynthesis (no additional parameters are needed) and (b) it is supported both by controlled-environment studies (Smith and Keenan, 2020), including plants grown at low CO₂ (Harrison et al., 2021), and FACE experiments (Wang et al., 2017). We will add sentences in the Methods (Section 2.1) to make this explicit:*

The responses of photosynthetic properties to enhanced $CO_2$ as simulated by the P model have been validated against both Free Air Carbon dioxide Enrichment (FACE) experiments (Wang et al., 2017) and controlled-environment experiments (Smith and Keenan, 2020). Moreover, the model's implied response of photosynthetic capacity to $CO_2$ has been validated by measurements on plants experimentally grown at low (160 ppm) $CO_2$ (Harrison et al., 2021).

*We included discussion of the realism of the other EEO components used in the paper when describing each component in the Methods (Section 2.1) but we realise that the statement about the evaluation of the seasonal cycle of GPP was rather brief (lines 131-133), so we will modify this text to be more explicit:*

The model has been shown to capture observed LAI dynamics across all biomes at different temporal scales (weekly, seasonal, annual and interannual variability) both at individual eddy-covariance flux measurement sites and when compared to satellite-derived LAI (Zhou et al., 2025). Furthermore, it predicts both the multi-year average LAI and the annual trends in LAI better than the biosphere models used in the Trends and Drivers of Terrestrial Sources and Sinks of Carbon Dioxide (TRENDY) project (Zhou et al., 2025).

3) A direct comparison with existing models (e.g., DGVMs, LSMs, and ESMs) would strengthen the study's contribution. How does the EEO-based approach improve upon these models in simulating GPP and C3/C4 competition?

*We included an evaluation of the different EEO components in the Methods (section 2.1) and have expanded this (see response to point 2 above). In particular, we now make it clear (both in the Introduction and Section 2.1) that the EEO approach provides predictions of the seasonal cycle of LAI and its inter-annual variability that are better than the state-of-the-art biosphere models that participate in the TRENDY project. Since this comparison has been made in other papers, it does not seem necessary to include a direct comparison with these models here. However, we will add a paragraph in the discussion about the advantages of the EEO approach it terms of reduced uncertainty and highlighting the overall better performance of these models compared to existing models, as follows:*

We have used a sequence of EEO-based models to simulate GPP and the relative contribution of $C_3$ and $C_4$ plants to overall productivity. Haas et al. (2023) also used the P model to simulate GPP at the LGM. Other studies of past vegetation changes have used models that simulate

changes in past vegetation on the basis of the competition between PFTs. PFT-based models require key physiological parameters to be specified separately for each PFT. The EEO modelling approaches used here avoid this complexity, considerably reducing uncertainties due to model parameterisation (Harrison et al, 2021) while at the same time representing the key processes of photosynthesis and plant growth accurately (Wang et al., 2017; Smith et al., 2019; Jiang et al., 2020; Lavergne et al., 2020; Peng et al., 2020; Smith & Keenan, 2020; Wang et al., 2020; Xu et al., 2021; Zhu et al., 2022). Furthermore, they capture recent trends in vegetation growth more accurately than the land-surface models used to predict the terrestrial carbon cycle (Cai et al., 2025; Zhou et al., 2025).

Additional references

Lavergne, A., Voelker, S., Csank, A., Graven, H., de Boer, H.J., Daux, V., Robertson, I., Dorado-Liñan, I., Martınez-Sancho, E., Battipaglia, G. et al., 2020. Historical changes in the stomatal limitation of photosynthesis: empirical support for an optimality principle. New Phytologist 225: 2484–2497.

Xu, H., Wang, H., Prentice, I.C., Harrison, S.P., Wang, G., Sun, X., 2021. Predictability of leaf traits with climate and elevation: a case study in Gongga Mountain, China. Tree Physiology. doi: 10.1093/treephys/tpab003.

Wang, H., Atkin, O.K., Keenan, T.F., Smith, N.G., Wright, I.J., Bloomfield, K.J., Kattge, J., Reich, P.B., Prentice, I.C., 2020. Acclimation of leaf respiration consistent with optimal photosynthetic capacity. Global Change Biology 26: 2573–2583.

4) The study reports an LGM GPP estimate of 84 PgC, within the CMIP6/PMIP4 range of 61–109 PgC. There is large inter-model variability (spanning tens of PgC).

*The CMIP6/PMIP4 range of GPP is indeed large, reflecting differences in the input data between the simulations (e.g. the ice sheet configurations used) as well as model-dependent differences in simulated climate and vegetation. As we pointed out in the Discussion (lines 338 to 342), attempts to constrain simulated GPP using oxygen isotope data from ice cores show an equally large range, because of large uncertainties in estimates of ocean productivity as well as model-dependent differences. We are not claiming that the estimate of 84 Pg is necessarily correct – only that the relative contribution of different factors to this reduction in GPP shown by the decomposition should be reasonable. We chose the MPI model because it has been shown to reproduce simulated climate better than most of the other models in CMIP6/PMIP4 (see lines 396-399). We did not use the simulated vegetation from this experiment because the EEO approach applies independentlky of vegetation types. We will expand the discussion of the LGM reduction to clarify that, while the absolute magnitude is uncertain, the partitioning of the causes of the reduction are more likely to be robust:*

Thus, although there is a consensus that GPP was considerably lower at the LGM than during pre-industrial times, and this is consistent with pollen evidence for a very large reduction in tree cover over much of the world (Prentice et al., 2000; Williams, 2003; Pickett et al., 2004; Marchant et al., 2009), the absolute magnitude of this change is uncertain. Nevertheless, since the climate simulated by the MPI ESM has been shown to reproduce pollen-based climate reconstructions better than most other CMIP6/PMIP4 models (Kageyama et al., 2021) and we use robust EEO-models to estimate the change in GPP, the partitioning of the impacts of different factors in the simulated reduction of GPP is likely to be robust.

5) The authors conclude that $CO_2$ effects led to a 3 PgC reduction in GPP during MH, while climate changes contributed to a 2 PgC increase, yielding a net difference of only 1 PgC. Given the large uncertainty in model estimates, is this difference statistically significant? Could this conclusion be influenced by model structural biases or sensitivity to parameter choices?

*We are using a single model to derive climate inputs. The EEO-models used to estimate GPP have very few parameters (at least an order of magnitude less than most land surface or vegetation models) and the values of these parameters have been explicitly derived from observations and/or experiments. Thus, our estimates of the effect of different drivers to changes in GPP during the MH are not expected to be influenced by structural biases or sensitivity to parameters.*

*The changes between MH and pre-industrial times are small, but they are consistent with expectations: GPP is reduced by the lower $CO_2$ but increased by the generally warmer and wetter climate in the northern hemisphere. We will modify the text in the Discussion to acknowledge that the MH changes are small but that the partitioning is consistent:*

The simulated overall change in GPP in the MH compared to the PI is small (< 1 PgC). Nevertheless, the changes in response to individual drivers are consistent with expecations: changes in climate and PPFD had a positive impact on GPP while the reduction in $CO_2$ in the MH compared to the PI had a negative impact on GPP. The positive effect of climate on GPP in the MH reflects changes in precipitation in now semi-arid regions of the sub-tropics, as a result of the orbitally induced expansion of the northern hemisphere monsoons and the lengthening of the growing season in the northern mid- to high-latitudes (Brierley et al., 2020). These changes in climate are reflected in our simulations. The northern extratropics are the only region to show an overall increase in GPP compared to the pre-industrial (4%) when $CO_2$ effects are included, but regions influenced by monsoon expansion, such as the Sahel and parts of South and East Asia, also show a tendency to increased GPP due to the MH climate.

---

## Author Comment (AC2)

We thank the reviewer for their comments on the paper. Our response to the specific points is given below in *italics* and proposed changes to the text in blue.

Zhao et al. investigated the impacts of climate fluctuations and CO2-induced alterations on gross primary production (GPP) using an eco-evolutionary optimality (EEO) based modelling approach. Two contrasting periods are focused, including the Last Glacial Maximum (LGM) and the mid-Holocene (MH), and compared to pre-industrial conditions (PI). This study assessed the importance of CO2, climate change, and light on the GPP at the global scale and pixel level. I have a few concerns about the robustness and implications of this study.

There is a large model range of GPP at the LGM (61-109 PgC yr-1 or 40-110 PgC yr-1), which could be primarily attributed to uncertainties in the effects of tree cover, distribution of C4 plants, and/or effects of climate change and changing CO2. Thus, the uncertainties associated with climate change and CO2 may be much larger than or at least comparable to their effects on the GPP differences between LGM and PI estimated by this study (Figure 5). This substantial uncertainty could raise questions about the robustness and reliability of the results presented in this study.

*The large range of GPP from the CMIP6/PMIP4 simulations was an issue raised by the first reviewer. As we said in our previous response, this range reflects differences in the input data between the simulations (e.g. the ice sheet configurations used) as well as model-dependent differences in simulated climate and vegetation. As we pointed out in the Discussion (lines 338 to 342), attempts to constrain simulated GPP using oxygen isotope data from ice cores show an equally large range, because of large uncertainties in estimates of ocean productivity as well as model-dependent differences. We are not claiming that our EEO-based estimate of 84 Pg is necessarily correct – only that the relative contribution of different factors to this reduction in GPP shown by the decomposition should be reasonable. We chose the MPI model because it has been shown to reproduce simulated climate better than most of the other models in CMIP6/PMIP4 (see lines 396-399). We did not use the simulated vegetation from this experiment because the EEO approach applies independently of vegetation types. In response to reviewer 1 we proposed expanding the discussion of the LGM reduction to clarify that, while the absolute magnitude is uncertain, the partitioning of the causes of the reduction are more likely to be robust:*

Thus, although there is a consensus that GPP was considerably lower at the LGM than during pre-industrial times, and this is consistent with pollen evidence for a very large reduction in tree cover over much of the world (Prentice et al., 2000; Williams, 2003; Pickett et al., 2004; Marchant et al., 2009), the absolute magnitude of this change is uncertain. Nevertheless, since the climate simulated by the MPI ESM has been shown to reproduce pollen-based climate reconstructions better than most other CMIP6/PMIP4 models (Kageyama et al., 2021) and we use robust EEO-models to estimate the change in GPP, the partitioning of the impacts of different factors in the simulated reduction of GPP is likely to be robust.

This study emphasizes the impacts of CO2 on the GPP and vegetation dynamics during both LGM and MH. Different levels of CO2 during LGM, MH, and PI, cause various light-use efficiency and distribution of C4 plants, thereby inducing different GPP. Figure 5 shows the magnitudes of effects of CO2 on the GPP during LGM and MH, thus could be used to estimate the sensitivity of GPP to CO2 changes. Although it is CO2 sensitivity based on the

long-term changes, it would be meaningful and interesting to compared it with that based on the recent observations and simulations.

*This was an issue that was also raised by Reviewer 1. We have shown that the response of our model is consistent with both controlled-environment studies (Smith and Keenan, 2020), including plants grown at low $CO_2$ (Harrison et al., 2021), and FACE experiments (Wang et al., 2017). We have proposed to add text in the Methods (Section 2.1) to make this explicit:*

The responses of photosynthetic properties to enhanced $CO_2$ as simulated by the P model have been validated against both Free Air Carbon dioxide Enrichment (FACE) experiments (Wang et al., 2017) and controlled-environment experiments (Smith and Keenan, 2020). Moreover, the model's implied response of photosynthetic capacity to $CO_2$ has been validated by measurements on plants experimentally grown at low (160 ppm) $CO_2$ (Harrison et al., 2021).

The description of units is confusing. For example, the unit of global GPP is PgC yr-1 rather than PgC. In Table 3, the unit of the contribution of C3/C4 to GPP is gC m2 yr. It should be checked because gC m-2 yr-1 is more commonly used.

*We gave the unit as PgC since we were referring to the annual total, but we agree that this is confusing and so we will change this systematically in the text and the tables to $PgC\ yr^{-1}$. The unit for the contribution of C3 and C4 to GPP is a global average contribution and thus is indeed $gC\ m^{-2}\ yr^{-1}$. However, we will correct the typo in the units in this table.*

---

## Author Response (AR2)

We thank the reviewers for their suggestions to improve our manuscript. Our response to specific point are given below *in italics*, with revised text in blue.

Review 1

The statement, "Thus, our estimates of the effect of different drivers to changes in GPP during the MH are not expected to be influenced by structural biases or sensitivity to parameters." may not be entirely accurate, as the authors themselves acknowledge the use of an empirical soil water stress function to scale GPP (Lines 173–175). This indicates that model sensitivity to parameterization may still influence the results.

*We do use an empirical function for soil water stress in applying the P model. Stocker et al. (2020) tested this function by comparing the performance of the P model before and after its application against eddy-covariance flux measurement sites under different levels of aridity and shown using this function improves model performance. The model with this empirical function provides a good prediction of the observations on multiple time scales. The impact of water stress on photosynthesis will not be affected by running the model under MH conditions. $CO_2$ levels which would affect photosynthesis through water use efficiency are explicitly and mechanistically accounted for in the P model. Overall, the EEO framework only uses a very small number of empirical parameters, and these are well quantified from global observations. Stocker et al. (2020) also showed that the sensitivity to these parameters was small. Therefore, our statement that the effect of different drivers on GPP will not be influenced by structural biases or sensitivity to empirical parameters seems justifiable.*

*In response to comments by the second reviewer, however, we have substantially expanded the description of the EEO models in general, and the application of the soil-moisture stress function in particular, to make it clearer how these are applied in the current context (please see revised text in response to the second reviewer's comment below).*

In their response, the authors emphasize that the novelty of this work lies in applying a parameter-sparse model (EEO) to investigate the relatively understudied mid-Holocene (MH) period. While this approach may offer advantages, the manuscript would benefit from a clearer explanation of how the EEO model has been validated. Specifically, the authors should provide evidence of model validation using independent proxies such as pollen reconstructions or other observational datasets. Since the conclusions draw upon the effects of $CO_2$ concentrations, precipitation, radiation, and C3/C4 vegetation fractions, the validation of the model's ability to simulate these drivers should be explicitly addressed. This could be presented as a table summarizing existing validations or as a standalone section detailing independent assessments.

*We provided information about the validation of each component of the model in the revised text, and have further expanded on this in the revised of Section 2.1 required by the second reviewer. This included validation of the impact of $CO_2$ under both enhanced and lower $CO_2$ levels. The performance of the models with respect to variations in climate parameters was also discussed in that Section. Since we have cited the appropriate evaluations in the text, the value of adding a table is unclear – although we could include the table below if this would make things clearer.*

*Our ability to validate the mid-Holocene and LGM simulations is limited. There is no global data source that provides information on GPP or $C_3$/$C_4$ vegetation fraction. As we stated in the Discussion, pollen data do not provide a way to separate $C_3$ and $C_4$ grasses. It could be used to estimate tree cover fraction, although this involves models such as REVEALS with their own methodological uncertainties and therefore would not provide a strong validation*

*of our results. There are some local or regional studies which provide compound-specific*
*δ¹³C analyses of leaf wax biomarkers. As we state in the Discussion, these regional studies*
*are consistent with our simulations of C₃/C₄, both in regions where C₄ was more abundant at*
*the LGM and the more limited regions where it was less abundant - this provides strong*
*support for our results. We have revised the text in the Discussion to make the palaeodata*
*limitations for global validation clearer and to point out the implications of the consistency*
*with the limited amount of regional data that exists as follows:*

[revised manuscript text omitted]

Additionally, my original concern regarding the large inter-model spread in Last Glacial Maximum (LGM) GPP estimates was not fully addressed. It remains unclear how the newly derived estimate fits within the broader context of previous estimates, and what implications this has for reducing uncertainty or reconciling existing model disagreements. A more in-depth discussion of how the new estimate compares to the existing range—and whether it provides any constraints or insights—would significantly strengthen the manuscript.

*We reiterate the point that we made last time about the difficulties of diagnosing the cause of the large range in previous estimates of GPP at the LGM. The forcing data used for these simulations (e.g. ice sheet configuration) varied, the climate models themselves differ structurally and in terms of specific parameter values used, and the vegetation component of these models are also different. Moreover, the vegetation components of the models are very complex being plant-functional type (PFT) based, and this involves specifying many (poorly constrained) parameters for each PFT. Evaluation of the performance of these models tends to focus on outcomes (e.g. vegetation distribution) rather than the individual processes (e.g. photosynthesis, evapotrainspiration) that gave rise to these outcomes. Our EEO-based models are parameter sparse, have been evaluated at process level using a wide range of different types of observation (controlled experiments, FACE experiments, field studies and geographic patterns of key traits in response to specific environmental gradients)  and therefore provides robust simulations of key vegetation properties. The EEO-based models have repeatedly been shown to out-perform the models that are routinely used to make projections in simulating the modern GPP. Although we are forced to use a climate model to provide LGM climate inputs, we have chosen the best-performing model as independently evaluated using quantitative climate reconstructions. Thus, we believe that our LGM simulations are likely to be more reliable than the previous estimates, although we have acknowledged in the Discussion that it would have been useful to run simulations with other CMIP6/PMIP4 models – but unfortunately other models did not archive all of the required variables. Since we recognise that we have perhaps been somewhat cautious about claiming our model is better tested and likely to provide more robust estimates of past GPP, we will re-organise and modify the Discussion section on the comparison with previous estimates, as follows:*

We have shown that the LGM was characterised by a large reduction in modelled GPP, while the mid-Holocene was characterised by a small increase in GPP compared to the pre-industrial state. Estimated GPP at the LGM was ca 84 PgC yr$^{-1}$ compared to ca 110 PgC yr$^{-1}$ in the PI. The simulated reduction at the LGM is consistent with previous model-based estimates (e.g. François et al., 1998; Prentice et al., 2011; Hoogakker et al., 2016), including those from the latest phase of the Couple Model Intercomparison project (CMIP6/PMIP4: Supplementary Table 3). However, previous estimates of GPP span a considerable range, from 40-110 PgC yr$^{-1}$). This reflects differences in the boundary conditions used, differences in the vegetation models used and their sensitivity to changes in CO2, and differences in the structure and parameterisations of the climate models overall. Diagnosing the specific causes of this large range is therefore extremely difficult. The parameter sparse nature of our EEO-based modelling approach, and the fact that the individual processes that give rise to the simulated GPP have been independently validated, suggest that our estimate of ca 84 PgC yr$^{-1}$ is more likely to be realistic than previous estimates. A limited number of studies have estimated GPP at the LGM by constraining model estimates using oxygen isotope records from ice core (Landais et al., 2007; Ciais et al., 2011; Yang et al., 2022). The still large range in simulated GPP (40-110 PgC yr$^{-1}$) reflects, in part, uncertainties associated with estimating ocean productivity and respiration fractionation rates. Thus, although there is a consensus that GPP was considerably lower at the LGM than during pre-industrial times, and this is consistent with pollen evidence for a very large reduction in tree cover over much of the world (Prentice et al., 2000; Williams, 2003; Pickett et al., 2004; Marchant et al., 2009), the absolute magnitude of this change is uncertain. Nevertheless, since the climate simulated by the MPI ESM has been shown to reproduce pollen-based climate reconstructions better than most other CMIP6/PMIP4 models (Kageyama et al., 2021) and we use robust EEO-models to estimate the overall change in GPP, the partitioning of the impacts of different factors in the simulated reduction of GPP is likely to be robust.

*We have also added a concluding sentence to the final paragraph of the Discussion to reiterate the advantages of using EEO-based models for the simulation of past vegetation, as follows:*

…. more accurately than the land-surface models used to predict the terrestrial carbon cycle (Cai et al., 2025; Zhou et al., 2025). Given their simplicity, the fact that the very few parameters required are well constrained from observations, and the demonstrated quality of their performance, EEO-modelling holds considerable promise for understanding past vegetation changes and their impact on climate.

**Review 2**

This is a very interesting study. Definitely, eco-evolutionary optimization modeling is an effective approach in estimating global GPP in different geological periods.
*We thank the reviewer for their positive evaluation of the worth of this approach*

The main issue for me is the lack of clarity regarding how the flowchart in Figure 1 was conducted in the simulations. While I understand the general intent, these steps raise important questions that I could not figure out and also could not find the answers in this manuscript and related references:
In Step 1: The use of the P-model to estimate potential GPP for C$_3$ and C$_4$ plants by assuming fAPAR = 1.0 is straightforward and aligns with standard applications of the P-model. This

part is clear and no problem for me.

In Step 2: The integration of global tree cover data and a C₃/C₄ model is where the confusion begins. It is unclear how tree cover data from the three specified periods (LGM, MH, and PI) were obtained, and more importantly, how these data were used in the C₃/C₄ model. Given that most C₄ species are grasses, the relationship between tree cover and the selection of C₃ versus C₄ GPP is not intuitive. Moreover, the referenced preprint describing the C₃/C₄ model lacks sufficient detail to explain it in this context. What mechanism or criteria does the model use to distinguish between C₃ and C₄ dominance? Without this, the reader cannot assess the validity of this step.

In Step 3: The introduction of the LAI model is also vague. No detailed descriptions are provided regarding the model's structure or rationale. The paragraph between lines 155–176 (Page 5) appears to describe this model, but it remains opaque after multiple times of reading. A clear explanation of how LAI and fAPAR are calculated and linked to the earlier steps is needed.

Step 4: The reapplication of the P-model and C₃/C₄ model using the updated fAPAR from Step 3 raises the question of whether more iterations are needed. Additionally, the use of the SPLASH model is introduced somewhat abruptly. It is said to provide a soil moisture correction to GPP (Lines 174–176), but the connection between cloud cover, soil moisture limitation, and GPP is not explained in sufficient detail.

Overall, the simulation system employed in this study appears to draw upon several existing models—P-model, C₃/C₄ model, LAI model, and SPLASH—some of which include optimization principles. However, the manuscript offers only general descriptions of these models and lacks a coherent and specific explanation of how they were implemented and integrated in this research and how the "optimal" was solved. Key model assumptions, parameterizations, and linkages between steps are either not provided or are difficult to follow. As a result, it is unclear how the full system operates or how optimization principles are realized within it.

I recommend that the authors substantially revise the methods section to clearly explain each step in the simulation workflow, including detailed roles of each model, data inputs, and how optimization is achieved. A more explicit and self-contained description would significantly improve the transparency and reproducibility of the study.

*We have expanded the model description to give a general overview of the simulation system we are using and the different steps involved, as shown in Figure 1. For each component model, we have provided a fuller description of the model, including the key assumptions and parameterizations. We have also revised the text to make it clear what EEO principles we are invoking in each of the models. We have also made it clear when we are using empirical relationships or empirically derived parameter values. The SPLASH model, which is not an EEO model, was only used to calculate soil moisture based on the climate model inputs, and this is then used to determine when to apply the soil moisture correction that is used in drier regions when running the P model. We have therefore expanded the description of this empirical correction and made it clear what the purpose of SPLASH is. The reapplication of the P-model and C₃/C₄ model in Step 4 using the updated fAPAR from Step 3 is in order to translate from the leaf level to the canopy level. This is therefore not an iterative process. We hope that the revised text (Section 2.1) explains what we have done more clearly. The revised text is as follows:*

[revised manuscript text omitted]

*We realised that we never actually specified that the tree cover information came from the MPI simulations, and have now added this information in section 2.2 – which we have renamed accordingly:*

**2.2. Derivation of LGM, MH and PI climate and tree cover inputs**

*The revised text to specify tree cover is:*

…… and uniquely has archived all the necessary climate and vegetation outputs needed to run the EEO-based models

The climate and tree cover outputs necessary to run the EEO-based models …….